# Optimized Nitrogen Fertilizer Rate Can Increase Yield and Nitrogen Use Efficiency for Open-Field Chinese Cabbage in Southwest China

Hailin Cao [1,2], Fen Zhang [1,2], Jian Fu [3], Xiao Ma [4], Junjie Wang [1,2], Fabo Liu [1,2], Guangzheng Guo [1,2], Yiming Tian [1,2], Tao Liang [1,2,5], Na Zhou [5], Yan Wang [5], Xinping Chen [1,2] and Xiaozhong Wang [1,2,*]

[1] College of Resources and Environment, Southwest University, Chongqing 400716, China; 18485766133@163.com (H.C.); cyx970726@email.swu.edu.cn (F.Z.); mahaoyue1@email.swu.edu.cn (J.W.); ai19970307@email.swu.edu.cn (F.L.); zhanjswu@email.swu.edu.cn (G.G.); tianym201005@163.com (Y.T.); zsh112233@email.swu.edu.cn (T.L.); chenxp2017@swu.edu.cn (X.C.)

[2] Interdisciplinary Research Center for Agriculture Green Development in Yangtze River Basin, Southwest University, Chongqing 400716, China

[3] Tongliang District Agricultural Technology Extension Service Center, Chongqing 402560, China; jianfu213@163.com

[4] Institute of Soil Science, Chinese Academy of Sciences, Nanjing 210018, China; mx1328@email.swu.edu.cn

[5] Chongqing Academy of Agriculture Sciences, Chongqing 400000, China; zn83692026@163.com (N.Z.); wangyan@163.com (Y.W.)

[*] Correspondence: wxz20181707@swu.edu.cn; Tel.: +86-1762-341-6568

**Abstract:** Intensive vegetable production has been characterized by high nitrogen (N) fertilizer input in southwest China. Optimizing the N fertilizer rate is the basis for the optimal management of regional N fertilizer. A two-year field experiment with five N fertilizer rates was conducted during 2019–2021 in southwest China, and the aim of this study was to identify the effects of different N application rates on yield, dry matter biomass (DMB), N uptake, N use efficiency (NUE) and soil mineral N ($N_{min}$) residues for Chinese cabbage (*Brassica chinensis* L.) and further determine the critical plant N concentration and root-zone soil $N_{min}$ residues required to reach the maximum DMB of Chinese cabbage. Five N treatments were established: control without N input (CK); optimal N fertilizer rate decreased by 30% (70% OPT, 175 kg N ha$^{-1}$), optimized N fertilizer rate (OPT, 250 kg N ha$^{-1}$), optimal N fertilizer rate increased by 30% (130% OPT, 325 kg N ha$^{-1}$) and farmers' N fertilizer practice (FP, 450 kg N ha$^{-1}$). The N source in all treatments was conventional urea (N ≥ 46.2%). The results showed that the total yield of Chinese cabbage followed a "linear-plateau" trend with an increasing N fertilizer rate. There was no significant difference in yield between the OPT, 130% OPT and FP treatments. The aboveground plant DMB and N uptake showed a 'slow-fast-slow' pattern with the growth period. There was no significant difference in aboveground plant DMB and N uptake between the OPT, 130% OPT and FP treatments. Moreover, the OPT treatment significantly increased the aboveground plant DMB and N accumulation by 29.6% and 40.5%, respectively, compared with the 70% OPT treatment. The OPT treatment significantly increased the NUE by 23.8%, 31.2% and 43.1% compared with that in the 70% OPT, 130% OPT and FP treatments, respectively. The linear-plateau model provided the best fit for the relationship among aboveground DMB of Chinese cabbage, plant N concentration and root-zone soil $N_{min}$ content. The critical root-zone soil $N_{min}$ and plant N concentrations were 94.1, 63.4 and 68.3 kg ha$^{-1}$ and 34.4, 33.5 and 32.9 g kg$^{-1}$ during the rosette, heading and harvest periods, respectively. In summary, compared to the FP treatment, the optimized N fertilizer rate (250 kg N ha$^{-1}$) could significantly reduce the N application rate, maintain yield, increase aboveground plant DMB and N uptake, and improve NUE. Moreover, the study has great significance for guiding the green utilization of vegetable N fertilizer in southwest China.

**Keywords:** Chinese cabbage; optimized N fertilizer rate; yield; N use efficiency; critical plant N concentration; critical root-zone soil $N_{min}$

## 1. Introduction

Vegetables play a critical role in human health because of their ample dietary fiber, vitamins and antioxidants [1,2]. China is the largest vegetable-producing country in the world and accounts for half of the total global vegetable production [3]. Southwest China is the dominant vegetable-producing area in China, accounting for 19.6% of the total cultivated area, and the total cultivated area and production in this region have increased by 126.1% and 163.4%, respectively, in the last 20 years [4]. However, intensive vegetable production in southwest China has been characterized by a high fertilization rate, especially for nitrogen (N). Research has shown that the average amount of N fertilizer applied for vegetable production in this region is 485.3 kg N ha$^{-1}$, 1.76 times higher than the crop nutrient requirement [5]. These excessive fertilizer inputs and high levels of rain lead to large reactive N losses, low fertilizer utilization rates and high environmental costs. Research has shown that N fertilizer utilization in pepper production in the region is less than 30% [6]. Therefore, there is an urgent need to optimize N fertilizer management for vegetable production in southwest China to achieve green vegetable production.

Optimizing the N fertilizer rate is the basis for optimal management of regional N fertilizer. In recent years, management strategies for systematic N fertilizer application in vegetable production have received increasing attention. Soil $N_{min}$, Kulturbegleitende-$N_{min}$-Sollwerte (KNS) and N-expert systems are commonly used in northwestern Europe [7]. The root-zone N management strategy can synchronize the soil N supply and plant N uptake in time and space [8], and this approach has been the most effective method for optimizing N application in current crop systems. Many previous studies have shown that root-zone N management can maintain or increase vegetable yield and improve N use efficiency through a reduction in the N fertilizer rate. One study showed that compared with conventional N application, a root-zone N management strategy reduced the N application rate by 53% and maintained cucumber yield [9]. Another study showed that a root-zone N management strategy for bitter gourd significantly increased the yield by 22.6% compared with conventional N management [10]. Moreover, a recent field trial showed that optimal fertilization based on a root-zone N management strategy could reduce N input by 37.5% and significantly increase NUE by 31.4% without sacrificing pepper yield [6]. Vegetable crops have different responses to N application rates across different regions due to large differences in soil and climate conditions and field management practices. For example, the optimal N application rates for tomato in east China and north China were 150 kg N ha$^{-1}$ in autumn–winter and 250 kg N ha$^{-1}$ in spring–summer [11,12]. In addition, the optimal N application rates were 180–196 kg N ha$^{-1}$ and 310 kg N ha$^{-1}$ for open-field radish [13] and bitter gourd [10] production systems, respectively. Therefore, more field experiments in different regions need to be conducted to explore the effects of N application rates on vegetable yield, plant N uptake and NUE and to further determine the optimal N application rates.

Establishing a relationship between critical soil $N_{min}$ content, plant N concentration and aboveground biomass is the key to defining the optimal N fertilizer rate [8,14]. The critical N concentration has been defined as the minimum N concentration required by the plant to obtain the maximum biomass [15]. In north China, the optimal soil $N_{min}$ content (0–90 cm) for high-yielding maize should be controlled within the range of 87–180 kg ha$^{-1}$ [14], and the critical soil $N_{min}$ content of high-yielding wheat is 50 kg ha$^{-1}$ during the jointing period (0–60 cm) and 87 kg ha$^{-1}$ during the harvesting period (0–90 cm) [16]. Compared with cereal crops, vegetables generally have a weaker nutrient absorption capacity due to shallow and sparse root systems [3,17,18]. The plant N concentration can accurately reflect the N nutrition status of crops and comprehensively explain the soil N supply capacity and crop N absorption capacity. Therefore, reaching the critical N concentration is a vital target value for adjusting the N application. Previous research has shown that the critical aboveground plant N concentration and maximum biomass of the wheat main stem in the jointing period were 2.42% and 0.7 g tiller$^{-1}$, respectively, and those in the flowering period were 1.62% and 1.95 g tiller$^{-1}$, respectively, which were similar to the values obtained from the observed

optimal N application treatment [15]. The relationship between plant N concentration and aboveground dry matter biomass (DMB) can be established to estimate the plant-critical N concentration more conveniently. These previous studies have focused on cereal crops, and more studies on critical plant N and soil $N_{min}$ concentrations should be conducted.

Chinese cabbage (*Brassica chinensis* L.) is the most frequently cultivated vegetable in southwest China. Therefore, using open-field Chinese cabbage (Fengkang 70, the most widely planted variety in this region) as the research object, we conducted a two-year field experiment to explore the effects of different N fertilizer rates on the growth of Chinese cabbage during 2019–2021 in southwest China. The objective of this study was to (1) indicate the effects of different N application rates on the yield, N uptake and NUE of Chinese cabbage to further identify the optimal N application rate and (2) determine the critical plant N concentration of maximum aboveground DMB and critical root-zone soil $N_{min}$ concentration during the critical growth periods of Chinese cabbage. Therefore, this study will provide theoretical support for the sustainable N production of open-field vegetables in southwest China.

## 2. Materials and Methods

### 2.1. Experimental Site

A two-season field experiment was conducted at the Hechuan Base of the Southwest University Experimental Farm (30°0′ N, 106°7′ E) from October 2019 to January 2020 and October 2020 to January 2021 in Chongqing Province, China. The crop planting system is a typical cabbage–pepper rotation system in southwest China. The region has a typical subtropical climate; summer is hot and rainy, and winter is mild and has less precipitation. The average temperatures were 9.9 °C and 11.3 °C, and the total precipitation was 76.2 mm and 134.9 mm, respectively, during the two seasons of the field experiment. The cultivated soil in this study was classified as purplish soil (Orthic Entisol), which developed from the fast physical weathering of sedimentary rocks of the Trias-Cretaceous system. Nowadays, those soils are widely distributed in the hilly areas along the Yangtze River in China, especially in the Sichuan Basin, which has an area of 300.0 thousand $km^2$ [19]. Formed in a grayish brown purple rock, generally, the coarse sand (2–0.2 mm), fine sand (0.2–0.02 mm), silt (0.02–0.002 mm) and clay (<0.002 mm) content in soil accounts for approximately 7.4%, 40.7%, 36.6% and 15.3%, respectively [20]. The main properties of the soil in the top 20 cm layer prior to the start of the experiment were as follows: pH, 5.65 (soil: water = 1: 2.5); total N, 0.50 g $kg^{-1}$; soil $NO_3^-$-N and $NH_4^+$-N, 4.89 and 2.06 mg $kg^{-1}$, respectively; available P, 19.5 mg $kg^{-1}$; exchangeable K, 56.0 mg $kg^{-1}$; and organic matter, 9.19 g $kg^{-1}$.

### 2.2. Experimental Treatments

The field experiment used a randomized complete block design with treatments of five N application rates and four replicate plots; each plot was 46.5 $m^2$ (5.6 m × 8.3 m) and included 240 Chinese cabbages. Natural precipitation in this region was high during the whole growth period of Chinese cabbage, and this high precipitation could basically meet the water demand of Chinese cabbage growth; therefore, we only needed to irrigate once through hose watering when Chinese cabbage was transplanted. The following N treatments were established: (1) control without N fertilizer input (CK); (2) optimal N application rate was decreased by 30% (70% OPT, 175 kg N $ha^{-1}$); (3) optimized N fertilizer rate (OPT, 250 kg N $ha^{-1}$) was based on the analysis of literature and expert recommendations; (4) optimal N application rate was increased by 30% (130% OPT, 325 kg N $ha^{-1}$); and (5) farmers' N application practice (FP, 450 kg N $ha^{-1}$), which was based on a survey of 52 farmers in southwest China. In this experiment, the selected N fertilizers were conventional urea (N ≥ 46.2%), superphosphate ($P_2O_5$ ≥ 12%), and potassium sulfate ($K_2O$ ≥ 52.0%) produced by Sichuan Tianhua, Shaanxi Hanzhong Tangfeng Chemical, and SDIC Xinjiang Lop Nur Potash Salt Co., Ltd. (Hami, China), respectively. More details on the fertilizer rate at different growth periods are listed in Table 1.

**Table 1.** Fertilizer application rates at different growth periods under different treatments (kg ha$^{-1}$).

| Treatment | Fertilizer Application Rates (N-P$_2$O$_5$-K$_2$O, kg ha$^{-1}$) | | | |
|---|---|---|---|---|
| | Seedling Period | Rosette Period | Heading Period | Total |
| CK | 0–60–145 | 0–30–72.5 | 0–30–72.5 | 0–120–290 |
| 70% OPT | 52.5–60–145 | 70–30–72.5 | 52.5–30–72.5 | 175–120–290 |
| OPT | 75–60–145 | 100–30–72.5 | 75–30–72.5 | 250–120–290 |
| 130% OPT | 97.5–60–145 | 130–30–72.5 | 97.5–30–72.5 | 325–120–290 |
| FP | 270–115–125 | 90–57.5–62.5 | 90–57.5–62.5 | 450–230–250 |

In the 2019–2020 experiment, Chinese cabbages were transplanted on 24 October, and basal and two topdressing fertilizer applications were applied in the seedling, rosette and heading periods on days 11, 40 and 64 after transplanting, respectively. In the 2020–2021 experiment, Chinese cabbages were transplanted on 10 October, and basal and two top-dressing fertilizers were provided during the seedling, rosette and heading periods on days 13, 49 and 73 after transplanting, respectively. The fertilizer application rates in the different periods for each treatment are shown in Table 1. All fertilizers were applied at a depth of 9–12 cm below the soil surface near the Chinese cabbage plants and then covered with soil. The first season of Chinese cabbage was harvested three times on 20 November and 14 December 2019, and 18 January 2020; The second season of Chinese cabbage will be harvested three times, on 16 November and 8 December 2020, and 12 January 2021. In both years, Chinese cabbage were cultivated with a row spacing of 60 cm and a within-row plant spacing of 40 cm. All other field management was conducted in accordance with local practices.

*2.3. Sample Collection and Analysis*

In two Chinese cabbage seasons, the root-zone soil (0–60 cm) was sampled with a soil drill through a "diagonal" five-point sampling method. Each sampling point was set at about 20 cm away from Chinese cabbage plants and divided into three soil layers (0–20 cm, 20–40 cm and 40–60 cm). Part of the soil samples were screened (2.00 mm sieve size) and air-dried for the determination of basic physical and chemical indexes. The other part of the soil sample was sieved and brought back to the laboratory; the $NH_3^+$-N and $NO_3^-$-N extracted by 0.01 mol L$^{-1}$ calcium chloride solution. The extraction solution was determined by Auto Analyzer 3 Continuous-flow Analysis-CFA (SEAL Analytical GmbH, Norderstedt, Germany).

We used a portable chlorophyll meter (SPAD-502, Minolta Camera Co., Ltd., Osaka, Japan) to measure SPAD values of 6 Chinese cabbage plants in each plot, and the average of 6 SPAD values was used as the SPAD value of the plot. In addition, 18 plants with uniform growth were selected from two rows in the middle of each treatment plot, and their total yield and commercial yield were measured. Samples of Chinese cabbage were collected, washed and placed in a constant temperature oven at 75 °C, dried to a constant weight, and then weighed. The dry samples of different organs were subsequently ground into a powder to determine the N concentration (Kjeldahl procedure). N accumulation was calculated using the formula DMB × $N_C$, where DMB and $N_C$ represent the aboveground dry matter biomass and the N concentration during the harvest period, respectively. The NUE index is expressed by the recovery efficiency of N (REN), agronomic efficiency of N (AEN) and partial factor productivity of N (PFPN) [6,13,21] and was calculated using the following formulas:

$$\text{REN (\%)} = (\text{plant N accumulation in N application treatment} - \text{plant N accumulation in CK treatment}) \div \text{N rate} \times 100.$$

$$\text{AEN (kg kg}^{-1}) = (\text{Yield in N application treatment} - \text{Yield in CK treatment}) \div \text{N rate}.$$

$$PFPN \ (kg \ kg^{-1}) = Yield \ in \ N \ application \ treatment \div N \ rate.$$

### 2.4. Statistical Analysis

IBM SPSS Statistics 27 (IBM Co., Ltd., New York, NY, USA) software was used to perform a two-factor analysis of variance on the data obtained from different N treatments and two growing seasons. At a probability level of 0.05, minimum significance (LSD) was used to compare the differences between treatments. Multiple comparisons of Tukey's test were performed to evaluate changes in treatment and year. The chart was created by Origin 2022 (OriginLab, Hampton, MA, USA). The relationships between yield and N application rates, aboveground DMB and plant N concentration, plant N concentration and soil $N_{min}$, and aboveground DMB and soil $N_{min}$ of Chinese cabbage were evaluated using SAS 9.4 (SAS Institute Inc., Cary, NC, USA) by linear regression, linear-plateau and quadratic models. The model with the highest $R^2$ was selected as the best fit for the relationship.

## 3. Results

### 3.1. Yield and Commodity Yield

The average total yields of Chinese cabbage in the CK, 70% OPT, OPT, 130% OPT and FP treatments were 46, 145, 179, 176 and 178 t ha$^{-1}$, respectively, during 2019–2021 (Figure 1). Compared to the 70% OPT treatment, the OPT, 130% OPT and FP treatments significantly increased the total yield by 23.4%, 21.4% and 22.8%, respectively ($p < 0.05$). There was no significant difference in the total yield between the OPT, 130% OPT and FP treatments (Figure 1, $p < 0.05$). The linear-plateau model best fit the relationship between the N application rates and the corresponding yield of Chinese cabbage, and the critical N application rate (the minimum N rate required to achieve the maximum yield) was 233 kg N ha$^{-1}$ during 2019–2021 (Figure 2). The two-year average commodity yields of the CK, 70% OPT, OPT, 130% OPT and FP treatments accounted for 41.3%, 72.9%, 75.3%, 74.4% and 73.8% of the total yield, respectively (Figure 1). The trend of the difference in commodity yield among the different treatments was similar to that of the total yield. Except for the CK treatment, there was no significant difference in the non-commodity yield of the other four N treatments ($p < 0.05$). There was no significant difference in the total yield of the two Chinese cabbage seasons for treatments with identical N rates.

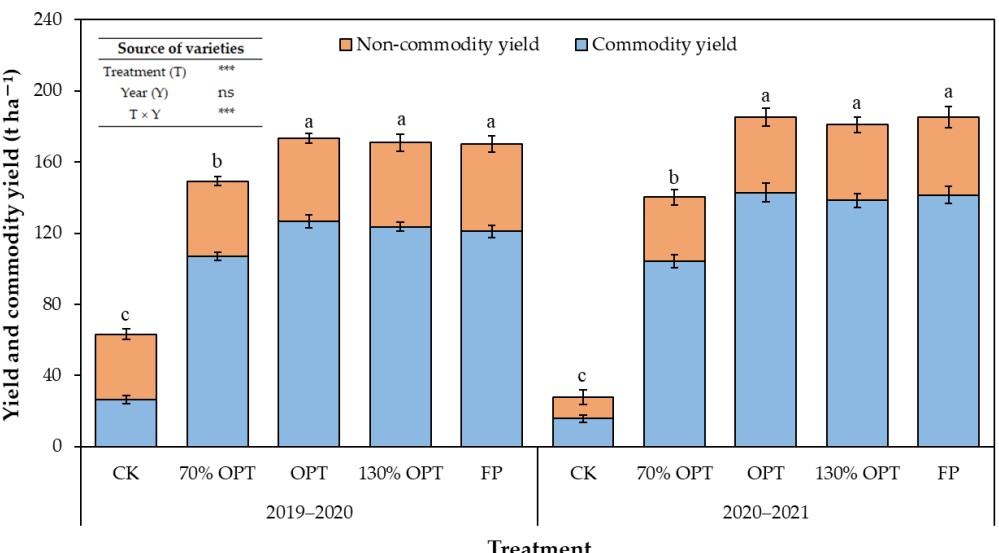

**Figure 1.** Effects of different N fertilizer application rates on the total yield and commodity yield of Chinese cabbage from 2019–2020 and 2020–2021. Different letters indicated that the total yield of Chinese cabbage was significantly different by Duncan's multiple comparison test ($p < 0.05$). ns, not significant. ***, significant at the $p < 0.001$ level.

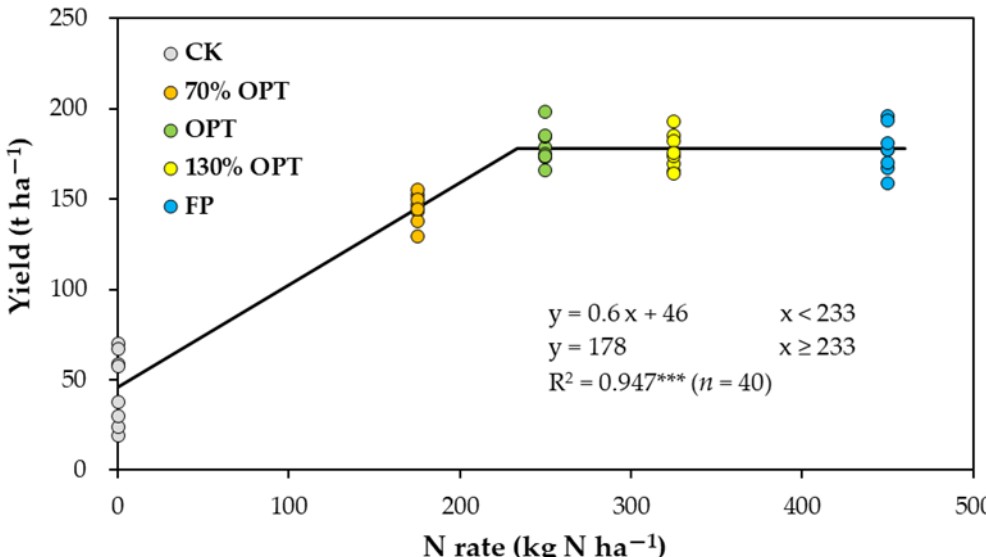

**Figure 2.** Relationship between yield and different N application rates for Chinese cabbage production during 2019–2021. ***, significant at the $p < 0.001$ level. A two-way ANOVA analysis found that although there was a significant interaction between treatment and year ($p < 0.001$) (Figure 1), this interaction was determined by the variance among treatments, and there was no significant difference between years.

### 3.2. Aboveground DMB

With the increase in the total yield of Chinese cabbage during the growth periods, the dynamic accumulation of dry matter biomass (DMB) showed a "slow-fast-slow" trend (Figure 3). Compared with CK, the four N treatments significantly increased the DMB accumulation of Chinese cabbage ($p < 0.001$). Over the two cropping seasons, compared with the 70% OPT treatment, the plant DMB accumulation of the OPT, 130% and FP treatments significantly increased by 57.3%, 54.6% and 50.9% during the rosette period; 30.6%, 21.0% and 21.8% during the heading period; and 29.6%, 28.2% and 28.9% during the harvest period, respectively ($p < 0.001$). Moreover, there was no significant difference in the two-year average DMB accumulation among the OPT, 130% OPT and FP treatments (Figure 3, $p < 0.001$).

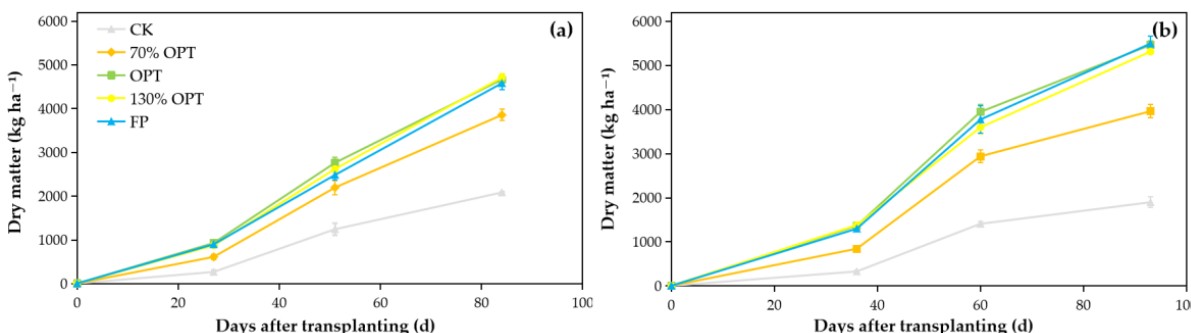

**Figure 3.** Effects of different N fertilizer application rates on dry matter biomass accumulation in Chinese cabbage from 2019–2020 (**a**) and 2020–2021 (**b**). The vertical bars represent the standard errors ($n = 4$).

### 3.3. Plant N Absorption and Accumulation

Over the two cropping seasons, compared with the CK treatment, the plant N concentration (Table 2) and N accumulation (Figure 4) of the four N application treatments increased significantly ($p < 0.001$). The aboveground plant N dynamic accumulation in Chinese cabbage was similar to DMB accumulation, showing an S-shaped trend (i.e., "slow-fast-

slow") (Figure 4). Compared with the 70% OPT treatment, the average aboveground plant N concentration of the other three N application treatments increased significantly (Table 2), and the average total N accumulation in the harvest period from the OPT, 130% OPT and FP treatments significantly increased by 40.5%, 39.6% and 43.1%, respectively (Figure 4, $p < 0.001$). Moreover, during the three key growth periods, there were no significant differences observed among plant N absorption and accumulation in the OPT, 130% OPT and FP treatments ($p < 0.001$). The interaction between year and treatment had no significant effect on plant N concentration in the three key growth periods (Table 2).

**Table 2.** The aboveground plant N concentration as affected by N treatments and year during three key growth periods: rosette, heading and harvest.

| Year | Treatment [1] | Aboveground Plant N Concentration (g kg$^{-1}$) | | |
|---|---|---|---|---|
| | | Rosette Period | Heading Period | Harvest Period |
| 2019–2020 | CK | 30.9 ± 0.3 c | 21.8 ± 1.2 b | 23.8 ± 0.1 c |
| | 70% OPT | 34.9 ± 0.8 b | 33.7 ± 0.2 a | 33.2 ± 0.5 b |
| | OPT | 37.0 ± 1.0 ab | 35.5 ± 0.3 a | 35.6 ± 0.5 a |
| | 130% OPT | 37.3 ± 1.4 ab | 35.8 ± 0.7 a | 35.3 ± 0.8 a |
| | FP | 38.0 ± 0.6 a | 35.9 ± 1.3 a | 36.5 ± 0.5 a |
| 2020–2021 | CK | 24.8 ± 0.7 d | 21.6 ± 1.1 c | 20.4 ± 0.4 c |
| | 70% OPT | 31.6 ± 0.3 c | 28.5 ± 0.6 b | 28.2 ± 0.5 b |
| | OPT | 35.4 ± 0.3 ab | 32.1 ± 0.4 a | 31.3 ± 0.2 a |
| | 130% OPT | 36.2 ± 1.0 a | 33.0 ± 0.6 a | 31.7 ± 0.4 a |
| | FP | 34.1 ± 0.7 b | 33.1 ± 0.5 a | 32.1 ± 0.5 a |
| | | Source of varieties | | |
| | Treatment(T) | *** | *** | *** |
| | Year(Y) | *** | *** | *** |
| | T × Y | ns [2] | ns | ns |

[1] The N application rate treatments were described in Table 1. [2] ns, not significant. ***, significant at the $p < 0.001$ level. Values represent the mean of four replicates. Within a column in the same cropping season, different letters denote a significant difference at $p < 0.05$ by Duncan's multiple comparison test.

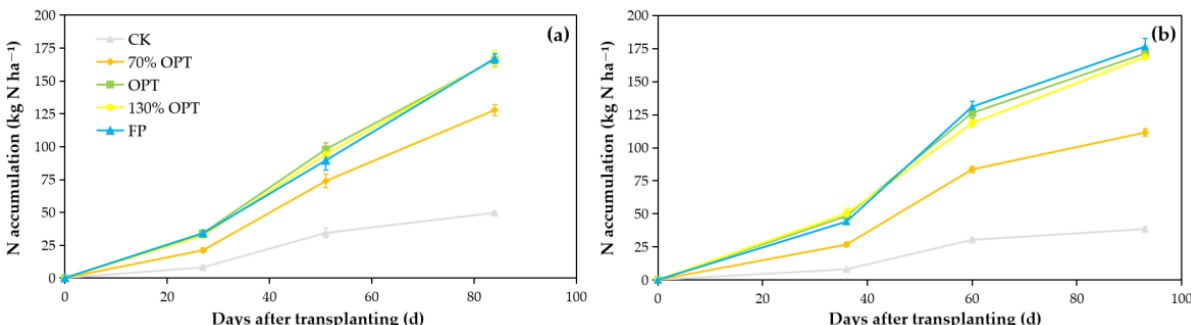

**Figure 4.** Effects of different N application rates on the dynamic N accumulation of Chinese cabbage during different growth periods from 2019–2020 (**a**) and 2020–2021 (**b**).

### 3.4. SPAD Values

N fertilizer application treatments significantly increased the SPAD values of Chinese cabbage compared with the CK treatment (Table 3). The average SPAD value in the harvest period from the OPT, 130% OPT and FP treatments significantly increased by 7.3%, 6.6% and 7.1%, respectively, compared with the 70% OPT treatment (Table 3, $p < 0.05$). In addition, the interaction between year and treatment significantly affected the SPAD value of Chinese cabbage during each growth period (Table 3, $p < 0.05$), indicating that the effect of the N application rate on the SPAD value varied with year.

**Table 3.** SPAD values at different growth periods under different treatments.

| Year | Treatment [1] | SPAD Values | | |
|---|---|---|---|---|
| | | Rosette Period | Heading Period | Harvest Period |
| 2019–2020 | CK | 26.3 ± 0.3 c | 25.2 ± 0.6 c | 26.8 ± 0.3 c |
| | 70% OPT | 28.6 ± 0.6 b | 30.1 ± 0.3 b | 30.8 ± 0.4 b |
| | OPT | 30.9 ± 0.2 a | 33.0 ± 0.3 a | 32.6 ± 0.2 a |
| | 130% OPT | 30.7 ± 0.4 a | 32.1 ± 0.5 a | 32.4 ± 0.2 a |
| | FP | 31.4 ± 0.4 a | 33.1 ± 0.3 a | 32.8 ± 0.4 a |
| 2020–2021 | CK | 27.0 ± 1.2 c | 27.9 ± 1.1 c | 25.2 ± 0.7 c |
| | 70% OPT | 37.0 ± 0.4 b | 38.7 ± 0.5 b | 31.2 ± 0.5 b |
| | OPT | 39.1 ± 0.3 a | 41.3 ± 0.5 a | 33.9 ± 0.3 a |
| | 130% OPT | 39.9 ± 0.5 a | 41.0 ± 0.5 a | 33.7 ± 0.4 a |
| | FP | 40.1 ± 0.5 a | 42.4 ± 0.5 a | 33.6 ± 0.5 a |
| | | Source of varieties | | |
| Treatment(T) | | *** | *** | ns [2] |
| Year(Y) | | *** | *** | *** |
| T × Y | | *** | *** | * |

[1] The N application rate treatments were described in Table 1. [2] ns, not significant. * and ***, significant at the $p < 0.05$ and $p < 0.001$ levels, respectively. Values represent the mean of four replicates. Within a column in the same cropping season, different letters denote a significant difference at $p < 0.05$ by Duncan's multiple comparison test.

### 3.5. NUE

Over the two seasons, the average maximum REN was 49.7%, which occurred in the OPT treatment (Table 4). Compared with the FP treatments, the OPT treatment significantly increased N recovery (REN) by 41.3%, agronomic N efficiency (NAE) by 82.3% and partial productivity of N fertilizer (PFPN) by 81.5% (Table 4, $p < 0.05$). Compared with the 130% OPT treatment, the OPT treatment significantly increased the REN, NAE and PFPN by 31.2%, 33.3% and 32.4%, respectively (Table 4, $p < 0.05$). In addition, compared with the 70% OPT treatment, the OPT treatment significantly increased the REN by 38.9%. However, compared with the 70% OPT treatment, the OPT treatment decreased AEN by 5.6% and PFPN by 13.3% (Table 4, $p < 0.05$). The interaction between year and treatment had no significant effect on REN, NAE and PFPN (Table 2).

**Table 4.** Effects of different N application rates on REN, NAE and PFPN of Chinese cabbage during 2019–2020 and 2020–2021.

| Year | Treatment [1] | REN (%) | NAE (kg kg$^{-1}$) | PFPN (kg kg$^{-1}$) |
|---|---|---|---|---|
| 2019–2020 | 70% OPT | 44.5 ± 2.4 a | 493 ± 15 a | 853 ± 15 a |
| | OPT | 46.4 ± 2.0 a | 441 ± 11 b | 693 ± 11 b |
| | 130% OPT | 35.9 ± 1.9 b | 332 ± 15 c | 526 ± 15 c |
| | FP | 26.0 ± 0.8 c | 238 ± 10 d | 378 ± 10 d |
| 2020–2021 | 70% OPT | 41.4 ± 1.6 b | 641 ± 25a | 801 ± 25 a |
| | OPT | 52.9 ± 1.6 a | 629 ± 20 a | 741 ± 20 b |
| | 130% OPT | 39.8 ± 0.5 b | 471 ± 14 b | 557 ± 14 c |
| | FP | 30.5 ± 1.4 c | 349 ± 13 c | 412 ± 13 d |
| | | Source of varieties | | |
| Treatment(T) | | *** | *** | *** |
| Year(Y) | | *** | *** | *** |
| T × Y | | ns [2] | ns | ns |

[1] The N application rate treatments were described in Table 1. [2] ns, not significant. ***, significant at the $p < 0.001$ level. Values represent the mean of four replicates. Within a column in the same cropping season, different letters indicate a significant difference at $p < 0.05$ by Duncan's multiple comparison test. The REN is the recovery efficiency of N, NAE is the agronomic efficiency of N, and PFPN is the partial factor productivity of N.

*3.6. Residual Soil $N_{min}$*

The root-zone soil $N_{min}$ ($NH_4^+$–N, $NO_3^-$–N) content increased significantly with the N application rate (Table 5). The average root-zone soil $N_{min}$ of the OPT treatment during the rosette, heading and harvest periods was 93.2, 86 and 97.3 kg N $ha^{-1}$, respectively (Table 5). Compared with the FP treatment, the 0–60 cm root-zone soil $N_{min}$ of the OPT treatment was significantly decreased during the rosette, heading and harvest periods by 50.4%, 51.1% and 45.1%, respectively ($p < 0.05$). Compared with the 130% OPT treatment, the 0–60 cm root-zone soil $N_{min}$ of the OPT treatment was significantly decreased during the rosette, heading and harvest periods by 43.6%, 37.7% and 29.0%, respectively ($p < 0.05$). Moreover, the root-zone soil $N_{min}$ was not affected by the interaction between N treatment and year, except for the range of 40–60 cm during the heading period (Table 5, $p < 0.05$).

Over the two seasons, the linear-plateau model best fit the relationship between DMB and aboveground plant N concentration (or root-zone soil $N_{min}$), aboveground plant N concentration and root-zone soil $N_{min}$ of Chinese cabbage during the rosette, heading and harvest periods (Figure 5). The critical aboveground plant N concentrations were 34.4, 33.5 and 32.9 g $kg^{-1}$, and the corresponding maximum DMBs of Chinese cabbage plants were 1020, 3007 and 4869 kg $ha^{-1}$ during the rosette, heading and harvest periods, respectively (Figure 5a–c, $p < 0.01$). Compared with the CK treatment, the aboveground plant N concentration increased significantly with the root-zone soil $N_{min}$ ($p < 0.001$), the maximum critical N concentration was 36.3, 34.6 and 34.1 g $kg^{-1}$, and the corresponding critical root-zone soil $N_{min}$ reached 117, 106 and 108 kg N $ha^{-1}$ during the rosette, heading and harvest periods, respectively (Figure 5d–f). The values of critical N concentration and maximum DMB of the Chinese cabbage plants were similar to those obtained in the OPT treatment (Figures 3 and 5a–f, Table 2). Furthermore, the maximum plant DMB was basically maintained when the critical root-zone soil $N_{min}$ was in the rosette (94.1 kg $ha^{-1}$), heading (63.4 kg $ha^{-1}$) and harvest (68.3 kg $ha^{-1}$) periods (Figure 5g–i).

**Table 5.** Root-zone soil $N_{min}$ ($NH_4^+$–N, $NO_3^-$–N) contents as affected by N treatment and year during three different growth periods.

| Year | Treatment [1] | Soil Mineral $N_{min}$ (kg ha$^{-1}$) | | | | | | | | |
| | | Rosette Period | | | Heading Period | | | Harvest Period | | |
| | | 0–20 cm | 20–40 cm | 40–60 cm | 0–20 cm | 20–40 cm | 40–60 cm | 0–20 cm | 20–40 cm | 40–60 cm |
| | CK | 10.4 ± 2.6 d | 14.1 ± 0.8 c | 13.0 ± 0.8 e | 19.5 ± 2.5 c | 27.8 ± 0.3 e | 18.8 ± 1.4 c | 12.1 ± 1.4 b | 18.0 ± 0.7 c | 15.7 ± 1.7 d |
| | 70% OPT | 24.7 ± 3.2 c | 37.2 ± 7.4 b | 26.0 ± 2.7 d | 31.1 ± 5.9 b | 39.1 ± 9.3 d | 30.5 ± 3.5 b | 31.0 ± 2.2 a | 43.0 ± 6.1 c | 31.4 ± 3.6 c |
| 2019–2020 | OPT | 43.9 ± 2.8 b | 36.7 ± 4.0 b | 29.1 ± 2.8 c | 35.6 ± 3.4 b | 51.1 ± 5.6 c | 34.5 ± 8.2 b | 29.3 ± 1.7 a | 53.8 ± 1.4 b | 57.1 ± 7.2 b |
| | 130% OPT | 49.2 ± 11 b | 82.9 ± 19 a | 88.1 ± 18.3 b | 36.6 ± 4.0 b | 58.2 ± 10 b | 67.3 ± 5.1 a | 31.4 ± 4.7 a | 43.5 ± 3.6 c | 80.2 ± 14 a |
| | FP | 64.6 ± 16 a | 82.3 ± 17 a | 91 ± 13.3 a | 53.0 ± 3.5 a | 70.8 ± 3.3 a | 64.3 ± 4.3 a | 33.8 ± 2.3 a | 63.1 ± 7.6 a | 78.6 ± 19 a |
| | CK | 8.9 ± 0.4 d | 9.8 ± 0.7 d | 9.7 ± 1.1 d | 7.3 ± 0.1 c | 8.9 ± 0.3 d | 8.1 ± 0.1 d | 6.0 ± 0.6 e | 5.8 ± 0.5 c | 5.5 ± 0.1 d |
| | 70% OPT | 13.0 ± 1.0 c | 20.7 ± 1.9 c | 16.9 ± 0.4 c | 12.7 ± 4.4 b | 13.7 ± 3.6 c | 13.3 ± 3.2 c | 16.6 ± 4.6 d | 4.6 ± 0.2 c | 4.7 ± 0.3 d |
| 2020–2021 | OPT | 17.6 ± 3.4 b | 30.6 ± 7.6 b | 28.5 ± 4.6 b | 12.8 ± 1.7 b | 18.5 ± 3.2 c | 19.5 ± 5.5 c | 34.0 ± 6.8 c | 8.0 ± 1.5 c | 12.3 ± 2.9 c |
| | 130% OPT | 22.0 ± 1.1 a | 52.6 ± 2.3 a | 35.5 ± 3.5 b | 25.4 ± 5.7 a | 38.8 ± 6.3 b | 49.6 ± 11 b | 46.0 ± 11 b | 35.8 ± 6.6 b | 36.4 ± 2.5 b |
| | FP | 24.4 ± 4.4 a | 55.2 ± 3.8 a | 58.3 ± 9.1 a | 25.9 ± 3.2 a | 59.9 ± 7.5 a | 78.0 ± 7.5 a | 58.4 ± 14 a | 67.0 ± 12 a | 53.3 ± 6.7 a |
| | | Source of varieties | | | | | | | | |
| Treatment(T) | | *** | * | ns [2] | * | *** | ns | ns | *** | *** |
| Year(Y) | | *** | *** | *** | ns | *** | *** | *** | *** | *** |
| T × Y | | ns | ns | ns | ns | ns | * | ns | ns | ns |

[1] The N application rate treatments were described in Table 1. [2] ns, not significant. *, *** significant at the $p < 0.05$ and $p < 0.001$ levels, respectively. Values represent the mean of four replicates. Within a column in the same cropping season, different letters denote a significant difference at $p < 0.05$ by Duncan's multiple comparison test.

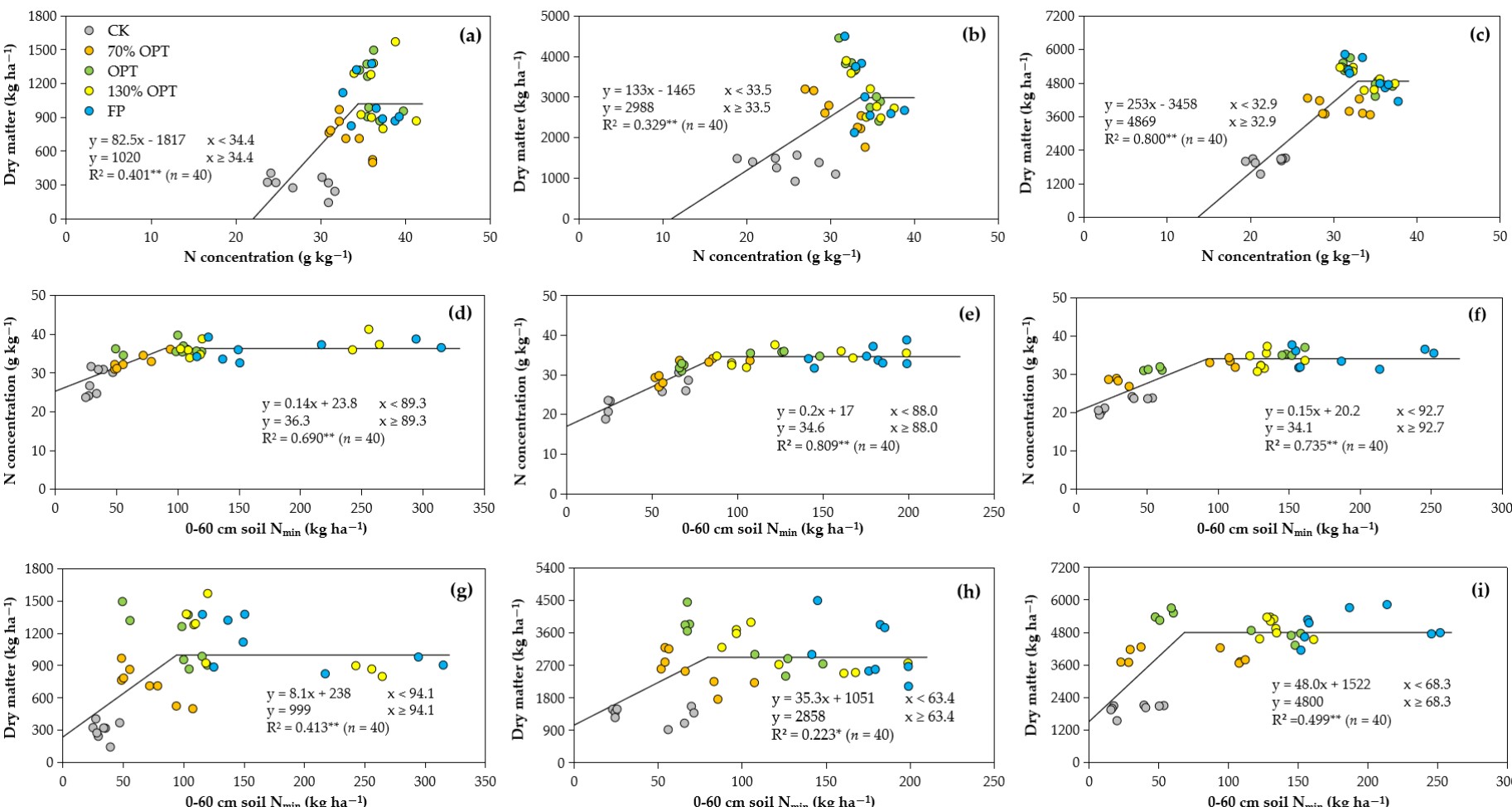

**Figure 5.** Across two growing seasons (2019–2021), the relationships between aboveground DMB and plant N concentration; aboveground plant N concentration and root-zone soil mineral N ($N_{min}$) at 0–60 cm depth; and aboveground DMB and root-zone soil $N_{min}$ at 0–60 cm depth during the rosette, heading and harvest periods were (**a**–**i**), respectively. Each data point represents the mean of four replicates for each N treatment of the current crop. *, ** significant difference at $p < 0.05$ and $p < 0.01$ levels, respectively.

## 4. Discussion

### 4.1. Response of Chinese Cabbage Yield to N Application Rates

Previous studies have shown that N fertilizer is a critical factor in increasing vegetable yield [10,13,22,23]. In this study, with the increase in the N application rate, the total yield of Chinese cabbage significantly increased by 215.2–289.1% (Figure 1). However, when the N application rate exceeded the nutrient requirement threshold of crops in the current season, crop yields did not increase significantly [13,15,24]. In this study, the relationship between the yield of Chinese cabbage and the N application rate was best fitted by a linear-plateau model (Figure 2), which was consistent with the findings of a previous study [15]. Therefore, in this study, we determined that the optimal N application rate was 250 kg N ha$^{-1}$. In addition, in this study, compared with the FP treatment, the OPT treatment reduced the N fertilizer input by 44.4% without sacrificing the Chinese cabbage yield. This is because local farmers invest more than half of the N fertilizer in the base fertilizer, and the excess N accumulates in the soil profile and is eluted into the deeper soil below the root zone over time [25], which results in insufficient nitrogen supply in the middle and late growth stages of Chinese cabbage and the inability of plant roots to capture sufficient nutrients from the soil, thus limiting the growth of Chinese cabbage. In contrast, in the OPT treatment, the N fertilizer inputs were adjusted based on the characteristics of Chinese cabbage nutrient demand in different growth periods, guaranteeing the root-zone soil N supply throughout the growth period and effective synchronization of plant N absorption [15] to maintain an ample yield of Chinese cabbage.

### 4.2. NUE

Currently, intensive vegetable production leads to low NUE due to excessive N fertilizer input; vegetables with NUEs lower than 40% are inefficient [13,26–28]. In this study, the NUE of Chinese cabbage under FP treatment was only 28.3% (Table 4), lower than that of cereal crops and vegetables in other countries [26,29,30]. This may be explained by two reasons. First, intensive vegetable production always has a large amount of N fertilizer input to obtain higher economic benefits, which is easy to cause soil acidification and has a feedback effect on vegetable NUE [31]. In addition, our experimental soil is a typical purplish soil in southwest China, which is characterized by a shallow soil layer and poor water and nutrient retention capacity, and the high temperature and rainfall throughout the year in this region caused high soil N leaching losses; these factors lead to a low initial soil N availability in the region [32]. A previous study found that crop NUE always decreased with increasing N application rate [13]. Similarly, in this study, the AEN and PFPN of Chinese cabbage decreased with an increasing N application rate (Table 4). However, in this study, we found that the REN of Chinese cabbage in the two seasons first increased and then decreased with increasing N fertilizer input. The OPT treatment significantly increased REN by 31.2% and 41.3% compared with the 130% OPT and FP treatments, respectively (Table 4). This is because the N application rate in the OPT treatment was reduced by 23.1% and 44.4%, and the yield of Chinese cabbage was maintained, compared to that in the 130% OPT and FP treatments. In addition, compared with the 70% OPT treatment, the OPT treatment significantly increased the REN by 38.9%. This may be because N in the 70% OPT treatment is not sufficient to completely supply plant nutrient requirements, leading to Chinese cabbage growth restrictions.

### 4.3. Root-Zone Soil $N_{min}$

Previous studies have shown that an appropriate amount of $N_{min}$ in the root zone is a key factor in improving crop yield [33–35]. In this study, over the two seasons, compared with the FP treatment, the average root-zone soil $N_{min}$ values for the OPT treatment during the rosette, heading and harvest periods were 93.2, 86 and 97.3 kg N ha$^{-1}$, respectively (Table 5), which basically satisfied the N nutrients required for the maximum aboveground DMB of Chinese cabbage plants at each key growth period. An excessive N fertilizer

application rate was a common FP according to the current farmers' practices; this approach leads to excessive residual soil $N_{min}$, with N gradually migrating to the deep soil with water (especially $NO_3^--N$) in the current or subsequent crop seasons [35–39], thus further increasing the risk of soil and groundwater pollution in vegetable plots. Therefore, the risk of N loss in vegetable plots could be significantly reduced by optimizing the N application rate.

*4.4. The Critical Plant N Concentration and Root-Zone Soil $N_{min}$ Were Determined*

In this study, the aboveground DMB of Chinese cabbage demonstrated a linear-plateau relationship with plant N concentration and root-zone soil $N_{min}$ (Figure 5a–c,g–i), similar to the findings of previous studies on food crops [15]. However, the plant N concentrations were inconsistent across crop systems. In this study, the critical aboveground plant N concentration for reaching the maximum aboveground DMB was 34.4, 33.5 and 32.9 g $kg^{-1}$ during the rosette, heading and harvest periods, respectively (Figure 5a–c), significantly higher than the values recorded by wheat in the North China Plain (27.2 g $kg^{-1}$ at the jointing stage and 15.3 g $kg^{-1}$ at the flowering stage) [15]. This may be due to the biological characteristics of the crops. Compared with cereal crops, vegetables have shallow roots but a large demand for nutrients [3]. Therefore, a higher critical plant N concentration was required to obtain a high yield. The critical root-zone $N_{min}$ concentrations for reaching the maximum DMB were 94.1, 68.4 and 63.1 kg $ha^{-1}$ during the rosette, heading and harvest periods, respectively (Figure 5g–i); these values are significantly lower than the critical N supply level of 150 kg N $ha^{-1}$ recommended by tomato [39] and the critical soil $N_{min}$ concentration of 200 kg N $ha^{-1}$ determined by in a cucumber–maize rotation experiment [9]. This is because tomatoes and cucumbers are grown in greenhouses, and generally, a higher nitrogen fertilizer is applied to meet the nutrients required for high plant yield [5]. In addition, when the N application rate reached 250 kg N $ha^{-1}$, the actual concentration of root-zone $N_{min}$ during each key growth period completely satisfied the N requirement of Chinese cabbage plants, which further verified that the root-zone N management strategy could synchronize the soil N supply and plant N uptake in the current season.

**5. Conclusions**

Two consecutive years of field experiments indicated that the OPT (250 kg N $ha^{-1}$) treatment maintained the yield of Chinese cabbage and plant N accumulation and increased the NUE by 41%, with a 44.4% reduction in the N fertilizer rate compared to the conventional farmers' practice in southwest China. These results indicate that appropriate N fertilizer rates at each key growth period are critical to synchronize soil N supply and vegetable crop demand and to achieve ample yield and a high NUE for Chinese cabbage. Furthermore, this study found that the N linear-plateau model best fit the relationship between aboveground plant DMB with root-zone $N_{min}$ and plant N concentrations of Chinese cabbage, and the critical root-zone soil $N_{min}$ and plant N concentrations were 94.1, 63.4 and 68.3 kg $ha^{-1}$ and 34.4, 33.5 and 32.9 g $kg^{-1}$ during the rosette, heading and harvest periods, respectively. These results are close to the actual values measured by the OPT treatment. Considering the evolution of environmental protection, vegetable ample yield and high NUE, the recommended N fertilizer rate was 250 kg N $ha^{-1}$ $season^{-1}$ of Chinese cabbage cultivation in southwest China. Our results show that southwest China has great potential for N reduction, and this study provides a theoretical basis for sustainable vegetable production in this region.

**Author Contributions:** Conceptualization, H.C., F.Z. and X.M.; methodology, H.C., F.Z. and X.M.; software, H.C. and F.Z.; validation, H.C. and F.Z.; investigation, H.C., F.Z., X.M., F.L., J.W., G.G., Y.T., J.F., T.L., N.Z. and Y.W.; data curation, H.C.; resources, H.C. and F.Z.; writing—original draft preparation, H.C. and F.Z.; writing—review and editing, X.W., H.C. and F.Z.; visualization, X.W., H.C. and F.Z.; supervision, X.W. and F.Z.; project administration, X.C. and X.W. All authors have read and agreed to the published version of the manuscript.

**Funding:** This research was financially supported by the National Natural Science Foundation of China (U20A2047); Cultivated land Quality Improvement in Chongqing during 2022—Investigation and evaluation of soil quality in vegetable base (4412200747); Innovation Research 2035 Pilot Plan of Southwest University (SWU-XDZD22001); and the Foundation of Graduate Research and Innovation in Chongqing under Project CYB21109.

**Data Availability Statement:** All data generated or analysed during this study are available from the corresponding author upon reasonable request.

**Conflicts of Interest:** The authors declare no conflict of interest.

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
