# Peer review of "Optimized Nitrogen Fertilizer Rate Can Increase Yield and Nitrogen Use Efficiency for Open-Field Chinese Cabbage in Southwest China"

_agronomy, doi:10.3390/agronomy13061578_

Round 1
Reviewer 1 Report
This manuscript improves upon basic fertility recommendations by evaluating root zone soil N in addition to traditional yield-based evaluation. The data presented is really quite good and does not show wide variation, which is sometimes associated with these types of trials. I have mostly minor comments that may help improve the manuscript, but I would vote to accept with minor revisions.
In the methods section the term "purple soil" is very familiar to scientists in China, but perhaps not in the rest of the world. Could the authors add additional notation on the type of soil (ie. Entisol) next to this. Also, do the authors have any information on the percent sand, silt and/or clay in the soil that can be added?
The plot dimensions are added, can the authors include the approximate number of plants per plot as well.
Can the authors include the variety of cabbage grown?
The plots were managed with local practices, specifically were they irrigated with sprinklers, drip, or furrow irrigation or not at all? Please add irrigation methods as it can impact nutrient leaching.
How were the plots sampled for the root zone nitrogen - how many subsamples per plot and so on.
Briefly add how SPAD measurements were done.
Can the authors add a line in the results section of yield that there was not an interaction between treatment and year, therefore allowing data to be pooled for figure 2? - Even if there was an interaction I am sure it would have been minor after looking at figure 1, but a short statement indicating this would be helpful.
English is very good on this manuscript, only very minor grammar suggestions
This sentence was confusing:
Compared with cereal crops, vegetables generally have stronger nutrient absorption capacity due to shallow and sparse root systems - ---Because vegetables have shallow and sparse root systems, wouldn't they have weaker nutrient absorption capacity compared to cereals?
Author Response
Response to Reviewer 1 Comments
Dear reviewer:
We appreciate your valuable comments and suggestions to improve the quality of our manuscript titled “Optimized nitrogen fertilizer rate can increase yield and nitrogen use efficiency for open-field Chinese cabbage in Southwest China” (No. agronomy-2409170). We have made significant efforts to incorporate the comments and revise the manuscript accordingly. The detailed changes in the revised version were highlighted in red color, and all details of changes and revisions were listed below for your reference.
Sincerely,
Xiaozhong Wang
Point 1: In the methods section the term "purplish soil" is very familiar to scientists in China, but perhaps not in the rest of the world. Could the authors add additional notation on the type of soil (ie. Entisol) next to this. Also, do the authors have any information on the percent sand, silt and/or clay in the soil that can be added?
Response 1: Thanks for your suggestion. The purplish soil is an Orthic Entisol, which have developed from the fast physical weathering of sedimentary rocks of the Trias-Cretaceous system. Nowadays, those soils widely distributed in the Sichuan Basin, which has an area of 300.0 thousand km². Formed in a grayish brown purple rock, generally, the coarse sand, fine sand, silt, and clay content in soil approximately accounted for 7.4%, 40.7%, 36.6%, and 15.3%, respectively. We have added the relevant information (lines 110-116) in the revised version.
Point 2: The plot dimensions are added, can the authors include the approximate number of plants per plot as well.
Response 2: Thanks. Each plot transplanted 240 Chinese cabbage plants. We have added the information (lines 121-122) in the revised version.
Point 3: Can the authors include the variety of cabbage grown?
Response 3: Thanks for your suggestion. We have added the variety of Chinese cabbage (Fengkang 70) in the revised version (lines 93-94).
Point 4: The plots were managed with local practices, specifically were they irrigated with sprinklers, drip, or furrow irrigation or not at all? Please add irrigation methods as it can impact nutrient leaching.
Response 4: Many thanks. Natural precipitation in this region was high during the whole growth period of Chinese cabbage, and this high precipitation could basically meet the water demand of Chinese cabbage growth, therefore we only need to irrigate once through hose watering when Chinese cabbage is transplanted. we have added the irrigation method (lines 122-124) in the revised version.
Point 5: How were the plots sampled for the root zone nitrogen - how many subsamples per plot and so on.
Response 5: Thanks. The soil (0-60 cm) was sampled with a soil drill through, and the "diagonal" five-point sampling method was adopted. Each sampling point was set at about 20 cm away from Chinese cabbage plants and divided into three soil layers (0-20 cm, 20-40 cm and 40-60 cm). Part of the soil samples were screened (2.00 mm sieve size) and air dried for the determination of basic physical and chemical indexes. The other part of the soil samples were sieved and brought back to the laboratory, the NH₃⁺-N and NO₃⁻-N extracted by 0.01 mol L⁻¹ calcium chloride solution. The extraction solution was determined by Auto Analyzer 3 Continuous-flow Analysis-CFA (SEAL Analytical GmbH). Those details have been added in lines 147-153.
Point 6: Briefly add how SPAD measurements were done.
Response 6: Thanks for your reminding. We have added this related SPAD measurements in lines 154-156.
Point 7: Can the authors add a line in the results section of yield that there was not an interaction between treatment and year, therefore allowing data to be pooled for figure 2? - Even if there was an interaction I am sure it would have been minor after looking at figure 1, but a short statement indicating this would be helpful.
Response 7: Thanks for your suggestion. We have added a two-way ANOVA to show interaction between treatment and year in Figure 1. Our results indicated that although there was significantly interaction between treatmentyield and year (P<0.001), this interaction was determined by the variance among treatments, and there was no significant difference between years (P>0.05). Meanwhile, we have added this statement in Figure 2 .
Reviewer 2 Report
The methodology is incomplete.
Discussion of the results at given times is restricted to comparisons with other papers. It is necessary to discuss the reasons for the results.

Author Response
Response to Reviewer 2 Comments
Dear reviewer:
We appreciate your valuable comments and suggestions to improve the quality of our manuscript titled “Optimized nitrogen fertilizer rate can increase yield and nitrogen use efficiency for open-field Chinese cabbage in Southwest China” (No. agronomy-2409170). We have made significant efforts to incorporate the comments and revise the manuscript accordingly. The detailed changes in the revised version were highlighted in red color, and all details of changes and revisions were listed below for your reference.
Sincerely,
Xiaozhong Wang
Point 1: IN THE METHODOLOGY THERE IS NOTHING ABOUT THE USE OF SPAD IN THE EXPERIMENTS. IT IS NECESSARY TO INSERT THIS AS THE ANALYZES WERE CARRIED OUT WITH SPAD.
Response 1: Thanks for your reminding. We used a portable chlorophyll meter (SPAD-502, Minolta Camera Co. Ltd., Japan ) to measure SPAD values of 6 Chinese cabbage plants in each plot, and the average of 6 SPAD values was used as the SPAD value of the plot. Those details have been added in lines 154-156.
Point 2: IN THE METHODOLOGY THERE IS NOTHING ABOUT SOIL RESIDUAL ANALYSIS. YOU MUST INSERT THIS PART.
Response 2: Thanks for your suggestion. The root-zone soil (0-60 cm) was sampled with a soil drill through, and the "diagonal" five-point sampling method was adopted. Each sampling point was set at about 20 cm away from Chinese cabbage plants and divided into three soil layers (0-20 cm, 20-40 cm and 40-60 cm). Part of the soil samples were screened (2.00 mm sieve size) and air dried for the determination of basic physical and chemical indexes. The other part of the soil samples were sieved and brought back to the laboratory, the NH₃⁺-N and NO₃⁻-N extracted by 0.01 mol L⁻¹ calcium chloride solution. The extraction solution was determined by Auto Analyzer 3 Continuous-flow Analysis-CFA (SEAL Analytical GmbH). Those details have been added in lines 147-153.
Point 3: ‘4.2.NUE’: THE DISCUSSION IS VERY BASED ON COMPARISONS WITH OTHER PAPERS. IT IS NECESSARY TO ACTUALLY DISCUSS THE REASONS FOR THE RESULTS. FOR THAT, THE AUTHORS SHOULD USE THE DYNAMICS OF N ABSORPTION, THE FUNCTIONS OF THIS NUTRIENT IN THE PLANT AMONG OTHER POSSIBLE REASONS...
Response 3: Many thanks. In this study, the NUE of Chinese cabbage under the FP treatment was only 28.3%. This lower NUE may be explained as followed two reasons. Firstly, intensive vegetable production always in this region has a large amount of N fertilizer input to obtain higher economic benefits. Secondly, our experimental soil was a typical purplish soil in southwest China, which was characterized by a shallow soil layer and poor water and nutrients retention capacity, and the high temperature and rainfall throughout the year in this region caused high soil N leaching losses, those factors would lead to a low initial soil N availability in the region.
Point 4: ‘5Conclusion’ :THESE INFORMATION REPEAT WHAT IS ALREADY REPORTED IN THE RESULTS.
Response 4: Thanks for your suggestion. We have revised the conclusion , the more details were listed in lines 350-363.